# Integrating Health into Local Plans: A Comparative Review of Health Requirements for Urban Development in Seven Local Planning Authorities in England

**DOI:** 10.3390/ijerph20054079

**Published:** 2023-02-24

**Authors:** Rosalie Callway, Anna Le Gouais, Emma L. Bird, Michael Chang, Judi Kidger

**Affiliations:** 1Population Health Sciences, Bristol Medical School, University of Bristol, 1–5 Whiteladies Road, Bristol BS8 1NU, UK; 2Centre for Public Health and Wellbeing, University of the West of England, Bristol BS16 1QY, UK; 3Department of Civil Engineering, University of Bristol, Bristol BS8 1TL, UK

**Keywords:** planning policy, determinants of health, comparative review, urban development, spatial planning

## Abstract

A local plan is a statutory policy document that supports urban development decisions across a local government area in England. Local plans are reported to need more specific requirements for development proposals regarding wider health determinants to address potential health outcomes and health inequalities. This study reviews the integration of Health in Local Plans of seven local planning authorities through documentary analysis methods. A review framework was formulated based on health and planning literature regarding local plans, health policy and determinants of health and dialogue with a local government partner. The findings identify opportunities to strengthen the consideration of Health in Local Plans, including ensuring that policies are informed by local health priorities and signpost national guidance, strengthening health-related requirements for developers (e.g., indoor air quality, fuel poverty and security of tenure) and improving implementation of requirements for developers (e.g., through adoption of health management plans and community ownership). The study identifies further research needs regarding how policies are interpreted by developers in practice, and on national guidance for Health Impact Assessment. It highlights the benefit of undertaking a comparative review, contrasting local plan policy language and identifying opportunities to share, adapt and strengthen planning requirements regarding health outcomes.

## 1. Introduction

The local plan is a statutory policy document required by planning legislation and the National Planning Policy Framework (NPPF) for England [1], where it should provide:


*“A positive vision for the future of each area; a framework for addressing housing needs and other economic, social and environmental priorities; and a platform for local people to shape their surroundings.”*
(para 15, chapter 3, NPPF [2].)

The local plan is the principle spatial planning mechanism through which local planning authorities (LPAs) seek to shape urban regeneration and other land use management. Local plans establish a long-term (typically 20 year) framework for managing land uses and are the basis on which planning consent decisions are made regarding residential, commercial, industrial and other infrastructural planning applications. They are expected to be reviewed every five years and each revision has to be approved by the national Planning Inspectorate before they can be formally adopted [3]. One in ten LPAs still need their local plans to be formally adopted, and over 40 local plans (out of a total 333) are more than five years old and require updating [4]. The UK government has called for all LPAs to have an up-to-date local plan in place by December 2023 [4]. Planning responsibilities vary somewhat according to different types of LPA: unitary, district and London borough councils are principally responsible for the production of local plans, alongside their powers of planning consent, and enforcement against unauthorised development and contravention of planning conditions [5]. Parish councils, town councils and neighbourhood forums can also produce neighbourhood plans outlining development priorities and requirements at a more local scale, in accordance with the Localism Act 2011 [5].

The promotion of health is established as a central principle for local plans by the NPPF, which states that “*Planning policies and decisions should aim to achieve healthy, inclusive and safe places*” (para 92, chapter 8). The NPPF goes on to outline various obligations for LPAs to improve the quality of planning decisions and urban development regarding health outcomes (Figure 1), including requirements to produce local health and a strategies to outline local health priorities in order to comply with the Health and Social Care Act 2012. Since 2013, unitary authorities and county councils (in the two tier county and district council system) in England have had public health as one of their administrative functions. Unitary and district councils have an urban planning function [6]. Although there are no regulatory obligations, local government public health teams and local health and wellbeing strategies should be referred to and inform local plans and planning decisions. The NPPF also includes numerous recommendations and voluntary requirements that are likely to affect health outcomes if implemented, including the use of design codes and the adoption of standard assessment frameworks, such as Building for a Healthy Life (para 133). These elements reflect the strong pathways of evidence about the links between the quality of the built environment and health outcomes [7,8,9,10].

Public Health England (PHE) and the Office for Health Improvements and Disparities (OHID), the agency that recently superseded PHE, have produced voluntary guidance that encourage the consideration and integration of health outcomes within spatial planning and policy [9]. This includes guidance for LPAs to conduct Health Impact Assessments (HIAs) in planning, which it describes as “*a process that identifies the health and wellbeing impacts (benefits and harms) of any plan or development project*.” [11]. PHE states that HIAs can be incorporated into wider Strategic Environmental Appraisals (SEAs) or Sustainability Appraisals, which are required to appraise local plans and planning policy, and larger development applications. HIAs can also be standalone exercises; however, as PHE notes, HIAs are not obligatory and it is left to the judgement of planners and developers whether and how an HIA is applied [11].

When preparing development proposals and planning applications, developers are expected to incorporate an array of national and local regulatory and planning policy requirements. Nationally, this includes the NPPF and other legislation, such as the Localism Act 2011 and The Town and Country Planning (Environmental Impact Assessment) Regulations 2017). They must also address local planning requirements and guidelines, as outlined in the local plan and supplementary planning documents [5]. A review of the planning system by the Town and Country Planning Association in 2018 stated that planning “*has undergone a bewildering rate of change and is now fragmented and confusing*.” [12]. Similarly, Barton and Grant (2013) noted that “*There have been recurrent calls for the planning system to employ new methods that will challenge well-established ways of working and better integrate the evaluation of potential outcomes of planning for more sustainable development”* [13]. Within this fragmented context, multiple studies have pointed to an absent systematic approach in the planning system towards health [14,15,16]. These studies refer to conflicting guidance, regulations and inadequate resources (at both national and local levels) as barriers to the effective implementation of planning decisions that would enable better health outcomes. The lack of regulatory and policy clarity regarding health expectations has implications for how health is reflected in development applications. A large study of different stakeholders involved in urban development (including 123 interviews with national government agencies, LPAs, developers, institutional investors, contractors, consultants and third sector actors) examined how health was considered in relation to urban development decisions [17]. Interview participants identified that planning policies were inconsistent or weak regarding how policies defined health requirements for developers when making planning applications. Interviews with developers also pointed to weak regulatory incentives for developers to take health into account when developing and implementing development proposals. Furthermore, some developers reported uncertainty about how different built environment forms would contribute to delivering specific health outcomes [17].

In a national survey of 175 public health and planning professionals, over a quarter of respondents did not agree with the statement that “*health is integrated into planning in my local authority*”, echoing this need to improve how health is reflected in planning policy [18]. The top three barriers to the integration of health identified by the survey related to translating local health evidence into practice, a lack of resources in local authorities to research health impacts and conflict between delivering sufficient quantity verses quality of development, where nationally required housing targets were often prioritised over the quality of homes [18]. Another identified problem regarding the integration of health concerns into planning policy relates to what is sometimes referred to as the “implementation gap”. This can occur when recommendations to apply evaluation frameworks such as Health Impact Assessments [13] do not result in changes in proposals or development delivery [19,20,21]. Another study found that, whilst local plans contained several issue-specific policies with implications for health (e.g., regarding transport infrastructure or housing), policies were not always clearly informed by local evidence regarding specific health challenges and they did not identify opportunities to promote health benefits derived from higher quality planning and development features [22].

This study therefore sought to examine and address four problem areas identified in the literature regarding local planning policies:Health is not systematically integrated and prioritised in local plans (including requirements to reflect local health and wellbeing strategic priorities and local health evidence);Local plans lack incentives for developers to prioritise health outcomes;Developers lack sufficient understanding about the form healthy urban development should take;Local plans lack requirements (including evaluation) to strengthen implementation by developers.

The study deepens the examination of these problems through developing a Health in Local Plans (HLP) review framework, in dialogue with OHID and an LPA partner, and applying the framework to a sample of seven local plans. The framework seeks to address the lack of systematisation of health requirements in local and national planning policies, through identifying specific evidence-based attributes and determinants of health that should be addressed through local plans. The work contributes to the field by seeking to establish a more coherent approach to considering health outcomes in future development proposals submitted and delivered within an LPA area. Learning from this study can support future application of the HLP review framework in the drafting of other local plans across England, and potentially more widely, to support the finer-grained integration of health priorities.

## 2. Methods

The following section outlines how the structure of the comparative review was developed, including involvement of planning policy practitioners, the structure of the review framework, the sampling approach for the local plans that are included in the comparative review and how the review was further refined and conducted. Figure 2 summarises the steps involved in formulating the framework.

### 2.1. Co-Production of the Health in Local Plans Review Framework

The study team included a Researcher-in-Residence who was embedded within a partner LPA (LPA 1), in support of the co-production and delivery of a range of system-based urban development and public health interventions [23]. The partner LPA was undertaking a review of their local plan in parallel with the study and had invited the Researcher-in-Residence and wider study team to present learning from other local plans to inform their local plan drafting process. This embedded relationship meant there was an opportunity to co-produce this study with local public health and planning policy practitioners to ensure the scope of the framework, and specific attributes were understood and seen as relevant to strengthening the health obligations for developers.

The main thematic areas and specific attributes of the HLP review framework were developed based on the foundation of Bird et al. (2018)’s paper, which consolidated systematic reviews of evidence regarding determinants of health and health outcomes in the built and natural environment (see Section 2.2.2 below). This initial framework was shared with the partner LPA officers. Their feedback regarding the particular attributes was considered and reflected in further iterations of the framework. None of the review attributes were removed as a result of dialogue with the LPA; rather, clarifications about definitions were made, to provide more specific criteria to apply when undertaking the review. Two additional determinants of health features, regarding fuel poverty and security of tenure, were added through the dialogue and a review of more recent literature post-dating the Bird et al. (2018) study. For certain features, the LPA officers were unsure of how realistic it was to address them through the local plan in terms of planning policy, rather than through development management requirements or building regulations, such as promoting the provision of healthy food environments in schools and retail outlets. The study team responded by looking at language adopted in the other sample local plans to illustrate whether and how it was possible to include that feature.

### 2.2. Structure of Health in Local Plans Review Framework

Building on the literature regarding Health in Local Plans, the HLP review framework and protocol for its application focuses on three areas:Definitions of health: How is health broadly defined and characterised in the local plan, including in terms of local strategic priorities and evidence?Health requirements for developers: What are the specific health-related requirements and determinants of health that developers are expected to address?Implementation of health requirements: How does the plan encourage delivery of the developer requirements?

#### 2.2.1. Review Area 1: Definitions of Health

The agreed attributes included for each area in the HLP review framework are discussed further below and shown in Appendix A Table A1. The first part of the review considered how health is broadly and specifically defined in the local plan. Here, the principle concern was whether the local plan policies are clearly informed by and communicate to developers local health priorities regarding distinct *preventable* health problems that the built environment and spatial layout have been evidenced to effect [7,8,9,16,24]. The review examined policy references to local health priorities, including (i) *non-communicable diseases*, such as type 2 diabetes, cancers, respiratory, heart and other diseases, and specific mental health concerns (including dementia, anxiety, depression); (ii) *health inequalities*, including differences in life expectancy rates within and between different demographics groups; and (iii) *planetary health* priorities regarding climate change and biological diversity, which have implications for the immediate and long-term health of humanity. Another attribute is the signposting of local health priorities, as identified by referring to the Health and Wellbeing Strategic priorities which all LPAs are legally required to produce, as well as identified through the local health evidence collected by health care trusts and LPA public health teams, and collated within key reports, such as Joint Strategic Needs Assessments and Indices of Multiple Deprivation. In particular, the Health and Wellbeing Strategy can provide an important framing for the local plan, clarifying local health priorities and how health should be addressed, including potential ways that development can contribute to mitigating negative impacts and improving health outcomes. The review also examined whether a Health Impact Assessment (HIA) was carried out on the local plans. An HIA can help to identify possible gaps and appraise high level health impacts of policies in the document [18]. Regarding standards and national guidance, by signposting or requiring developers to apply nationally recognised principles that seek to enhance development quality (e.g., Public Health England (PHE)’s guidance, “Spatial Planning for Health: An evidence resource for planning and designing healthier places” [9]), an LPA helps to give developers a clearer indication of the specific built environment or place attributes they are expected to incorporate. Such recommendations are important to give developers a clear indication of the baseline good practice that they are expected to refer to, regarding key built environment attributes that have peer-reviewed evidence of health benefits. It should be noted, however, that a planning requirement to seek certification of a voluntary standard does not necessarily guarantee the implementation and monitoring of those principles at the construction and in-use phases of a development [20,25,26]. As such, area three of the review (see Section 3.3) sought to address some of the key risks regarding this potential implementation gap.

#### 2.2.2. Review Area 2: Health Requirements for Developers

The second part of the review examined the health requirements for developers, regarding (i) the determinants of health, with scoring based on 5 built environment themes, 15 principles and 39 features with a clear pathway of evidence to particular health outcomes, based on the wider determinants of health framework established by Bird et al. (2018) (see Appendix A Table A2); (ii) a specific healthy development policy and requirements to conduct a Health Impact Assessment (HIA) for developments; (iii) planning verification checklists which refer to the HIA requirement; and (iv) specific determinants of health requirements of interest to the partner LPA, regarding fuel poverty and energy insecurity, outdoor and indoor air quality, secure tenure and affordable housing.

Regarding the determinants of health, this section of the review examines local plan policy references to 39 features identified as key social and environmental “determinants of health” in the built and natural environment based on work by Bird et al. (2018) [7,9]. Bird et al. categorised the features under five healthy development themes:Healthy neighbourhood design;Healthy housing;Healthier food environments;Natural and sustainable environments;Healthy transport.

Appendix A Table A2 outlines the determinants of health features included within these five themes and their reported health benefits in further detail. The HLP review framework also examined the specific health-related policies that developers are expected or encouraged to consider when making planning applications. This section recognised the “Health in All Policies” model promoted by the Local Government Association [27,28] by seeking to understand how an LPA uses policies to evaluate the quality of development in terms of the health priorities and targeted outcomes. The review includes whether the local plan has a specific healthy development policy, as well as whether a Health Impact Assessment is required by developers when preparing a proposal.

In addition, certain determinants of health were identified as specific local concerns for the partner LPA, and/or were the subject of policies that were being updated or introduced in the case study area, notably (i) fuel poverty and energy security, (ii) outdoor and indoor air quality, (iii) secure tenure and (iv) affordable housing policy provisions. First, in terms of *fuel poverty and energy insecurity*, the links with household provision of energy and health outcomes are also numerous. Energy efficiency was already identified as an important determinant of health in the HLP review framework, as efficiency measures (in combination with effective ventilation) are associated with reduced morbidities related to exposure to excess cold or heat, reduced respiratory and allergic symptoms, e.g., asthma and eczema, reduced sick building syndrome (SBS), as well as reduced blood pressure, sinusitis and chronic obstructive pulmonary disease (COPD) [29]. More recent studies indicate that energy insecurity and fuel poverty, typically associated with low-income households, can increase heat or cold stress, undermine the quality of sleep, exacerbate arthritic, mobility issues and cardiovascular issues, and have detrimental mental health affects [30].

Second, regarding *air quality,* poor air quality is already included as a feature in the Bird et al. (2018) study, where the health impacts of poor indoor and outdoor air quality are well established, including the risk to people with chronic conditions such as asthma and heart disease, an increased risk of birth outcomes and infant mortality, damage to cognitive function and decreased physical activity among older adults [7]. Air pollution has been found to particularly affect communities living in more deprived areas of more diverse ethnic backgrounds situated closer to poor ambient air quality [31]. Third, *security of tenure* refers to concerns about the poor quality of rental properties and lack of adequate tenancy rights, including ensuring a minimum guaranteed length of tenancy, and the consequences for people’s health and wellbeing. The issue of secure tenure was not included as a determinant of health in the Bird et al. (2018) review, but more recent systematic reviews indicate that the security of tenure for tenants in rented accommodation is associated with a number of physical and mental health benefits [32,33]. The real estate sector has predicted strong growth in the private rental and build-to-rent market in England in the next decade [34,35], suggesting that LPAs need to be clear about developer and/or landlord responsibilities towards tenancy rights to encourage better health outcomes. Finally, regarding *affordable housing*, there is strong evidence that this is an important determinant of health, in terms of improved social, behavioural and mental health outcomes for residents [7]. Policies that enable increased affordable housing provision are vital to address both health inequalities and for health promotion, as a review of health equity in England (2020) pointed out:


*“While poor-quality and unaffordable housing damages health and worsens health inequalities, good-quality and affordable housing contributes to improving health and wellbeing and reducing inequalities.”*
(“The Marmot Review Ten Years On”, 2020 [31].)

#### 2.2.3. Review Area 3: Implementation of Policy Requirements

The third part of the HLP review examined how local plans support the implementation of policy requirements by developers. This area is necessary to address the potential “implementation gap” or “leaky bucket” of urban design quality [19], referring to the risk of developers compromising on policies and planning conditions in development projects after planning consent is achieved [19,20,36,37,38]. This area of the review sought to pinpoint what language local plans use to encourage developers to follow through with policy requirements, including those relating to health. This area includes five aspects concerning implementation, notably (i) financial resources, (ii) viability appraisal, (iii) planning and management, (iv) monitoring requirements and (v) community ownership (see also Appendix A Table A1). Regarding *financial resources*, new development and intensified sites will involve long-term management and maintenance costs, as well as incur an additional burden on public services and amenities, e.g., GP surgeries, transport infrastructure, schools and natural spaces (green/blue infrastructure) that are essential for protecting against harms and promoting health benefits [39]. As such, the HLP will examine policy requirements for developer financial contributions, through Section 106 agreements, Community Infrastructure Levy (CIL) and other funding mechanisms as important means to enable ongoing delivery of those policies. *Viability appraisal* is an assessment associated with the provision of affordable housing units, which seeks to ensure that a developer can still meet housing targets within predicted returns, project costs and target margins. The HLP review is looking at requirements for viability appraisal and negotiations to be transparent (e.g., using an open book assessment process) so that viability appraisal is not used as a means to justify reductions in health benefits associated with affordable housing and design quality requirements [40,41].

In terms of *planning, management and maintenance requirements*, obligations to prepare health management and maintenance plans are necessary to encourage longer-term processes in place regarding the delivery of planning requirements [19,20,42]. Similarly, regarding *monitoring requirements*, local plans should give clear expectations for accountability from developers to keep track of policy requirement delivery, including whether there is a need to refine or change practice based on the findings of that evaluation [20,37,38]. The HLP review looks for requirements to undertake post-occupancy evaluations, including in relation to the production of a health management plan.

In relation to *community ownership,* various reports describe the importance of community ownership and leadership in development processes to help improve the quality and sense of place, benefitting longer-term mental and physical health outcomes [43,44,45,46]. Some models of development are particularly highlighted as helping to promote ownership, such as community land trusts, cooperative, co-housing and self-build projects [47,48,49,50,51,52]. The Self-build and Custom Housebuilding Act (2015) requires authorities to keep a register of interest for self-build projects. As such, the review is looking for policies that promote community-led housing and development models, as well as signposting to the self-build register, as well as whether there is clear signposting of good practice for early and effective community engagement and dialogue in relation to proposals. Planning legislation requires all LPAs to produce a Statement of Community Involvement (SCI) outlining how they involve local actors in plan making (NPPF para 126-7). The SCI can also require developers to apply community engagement practices in preparing an application and present this in their planning application (typically in the Design and Access Statement—DAS), particularly for large-scale developments. The DAS presents how a developer engaged and responded to public views regarding how their proposal will impact on healthy life choices and the local environment, helping to promote greater community ownership of the proposal. Although LPAs’ SCI includes mainly “soft” recommendations and guidance regarding how to produce an SCI, it is assumed that developers/planning consultants will look at this guidance when conducting community engagement activities ahead of making planning applications. However, a report by the University of Reading found that SCI is “*currently an under-utilised and under-performing tool in the planning system*” [53]. The HLP review examined whether the local plan clearly points developers in towards any guidance and requirements regarding how developers should apply the SCI.

### 2.3. Sampling of Local Planning Authorities

A small purposive sample of local plans was identified from 333 LPAs in England [54] to which we applied the review framework. The comparison of this sample was necessary to support the initial formulation of the review framework, to test out and refine the attributes in the framework, and ensure the applicability and relevance of review attributes and features for different local plans. The comparison across the local plans also enabled the identification of specific policy areas that were missing or could be enhanced in the partner LPA local plan or in other local plans that did not address a particular attribute. It was recognised that a representative sample of local plans would be difficult, if not impossible, to achieve because there is significant variation between LPAs in England, in terms of the type of authority, demographic composition, political make-up, as well as rural, peri-urban, and urban contexts. Thus, a pragmatic purposive sample of local plans from seven LPAs was selected (see Table 1), based on the following criteria:Practitioner and literature recommendations of good practice examples for incorporating health [11,22,46];Local plans that were recently revised or are in the process of revision and therefore reflective of more recent national regulatory, policy and health-related requirements;Predominantly unitary or metropolitan borough councils were included to align with our partner LPA, which is a unitary council;Whether the local plan included HIA of the policy, including examples with and without an HIA.

As the study was seeking to support the enhancement of the partner LPA’s local plan, the sample sought to include local plans that were already perceived to positively incorporate health attributes. Expert practitioners were consulted, as well as a review of grey and academic literature [11,22] regarding which “good practice” local plans to include in the review. Practitioners included an OHID officer, public health and planning policy officers at the partner LPA, as well as English LPA public health officers, via an online community of practice for public health officers (the Knowledge Hub: https://khub.net/, accessed on 12 August 2022).

It was important to include more recently updated local plans so that they reflected the most current national policy and regulatory changes and were therefore more comparable with the partner LPA local plan. Similarly, the partner LPA was a unitary city authority, which has broader planning functions than a district council (including over transport and larger infrastructure, as well as public health), so to aid comparability predominantly urban unitary, metropolitan boroughs and district councils were identified. One rural unitary authority (LPA 3) was included to consider if there was a significant difference in scope for rural authorities. Finally, noting that a Health Impact Assessment (HIA) of a local plan is not a mandatory requirement, the sample included three LPAs that had or were in the process of applying an HIA to the local plan and four that had not to examined whether the process of undertaking an HIA had an effect on the inclusion of different health attributes.

The resultant sample included a range of LPAs whose populations experience various levels of health and deprivation according to national rankings. Although these rankings were not taken into account in the selection of the seven local plans, the UK Indices of Multiple Deprivation (IMD) from 2019 indicates that the local areas included in this study present a range of low to middle IMD rankings (in other words more deprived) in comparison with the other English LPAs.

### 2.4. Conducting and Refining the Health in Local Plan Review

One member of the research team undertook the initial review. A word search was applied, alongside more detailed reading of the policies and general references of the local plan documents. This was to check content through searching for key terms, while also reading the text in depth to understand the meaning and context. In appraising each local plan, the reviewer considered whether there is an explicit policy, or requirement within a policy, which relates to a particular HLP attribute or particular determinant of health feature (as defined in Section 2 on health requirements for developers). Relevant policies were recorded in an Excel spreadsheet and a simple semi-quantitative score and traffic-light rating system applied to indicate the status of each attribute or determinant of health feature, where green (scoring 1) indicated that a HLP attribute or determinant of health feature was clearly defined in a policy, amber (scoring 0.5) indicated that an attribute or feature was partially defined (e.g., a brief reference but no specific requirements) and red (scoring 0) indicated that an attribute or feature was not present.

After the first researcher had completed the review of the local plans, a second researcher in the study team also conducted a “blind” review of LPA 1′s local plan. The purpose of the blind review process was to:To check that the review framework was understandable and applicable by other users and reviewers;To check that other reviewers produce a similar score using the framework, ensuring that there is a degree of consistency, and addressing potential areas of bias, such as selective attention bias or confirmation bias by the original reviewer;To identify areas for review and/or refinement: whether attributes should be added or removed, whether they could be better defined or require further explanation for both reviewers and other users.

The second reviewer produced similar review scores in comparison with the first reviewer when examining the partner LPA’s draft local plan against all three areas of the review framework (see Section 2.3). There was only a 1% difference in the scoring for specific determinant of health features (the second reviewer scored 31 points compared to 31.5 by the first reviewer, out of a possible 39). Regarding the wider HLP attributes, there was initially 88% agreement in scoring (for 15 out of 19 points). The second reviewer was unclear about the assessment of two attributes so did not score them. After the scoring and criteria definitions were compared and discussed, the dialogue helped identify certain attributes which needed clearer definition or specificity regarding what words to search for and the types of information that should be considered. Attribute and feature definitions were updated, with clarifications, for 7 of the 19 possible points.

## 3. Results

The following section outlines the findings from the comparative HLP review, focusing on the three review areas: definitions of health, health requirements for developers and implementation of developer requirements.

### 3.1. HLP Review Area 1: Definitions of Health

Table 2 provides an outline of the scores assigned to each definition attribute, relating to how each local plan refers to and is informed by specific health concerns (e.g., non-communicable diseases, mental health, health inequalities, planetary health); local health and wellbeing strategies; whether a Health Impact Assessment was applied to the local plan; local health evidence, e.g., Joint Strategic Needs Assessment; and signposts of national health-related standards, as well as health guidance and publications. The details for each item are described in more detail in Appendix A Table A1.

#### 3.1.1. Health Definitions: NCDs, Mental Health, Life Expectancy, Inequalities, Planetary Health

The amber scoring for all the LPAs regarding the “health definition” attribute in Table 2 indicates that the seven local plans referred to the broad concepts of health and wellbeing but did not identify particular local concerns and priorities in terms of local health inequalities. For example, the plans did not indicate or present maps of health inequalities at a ward or neighbourhood scale or report particular health trends over time. LPA 3′s local plan included some headline local health statistics regarding city-scale life expectancy levels, dementia and obesity levels in comparison with the national averages, but more detailed local health data were not highlighted in any of the local plans. Similarly regarding mental health, the local plans refer to broad concepts of mental health and wellbeing without presenting more localised and disaggregated concerns, relating to depression, anxiety, substance abuse and dementia. Two plans did provide area-wide headline statistics about the rates of dementia (LPA 3 and LPA 5). LPA 7 had a specific policy which referred to the impact of development of mental health care service provision. Finer-grained local information could help policy makers and developers to better understand the local health context for a particular area, and encourage them to consider how plans and development proposals are informed by and respond to that health context.

In terms of climate change requirements, relating to planetary health, all seven local plans included clear principles regarding climate mitigation and adaptation. Some local plans included climate change targets that developers are expected to contribute towards. Four local plans (LPA 2, LPA 4, LPA 5 and LPA 6) also established which sectors contribute most to local carbon baseline emissions. The partner LPA’s local plan did not report their principal baseline emissions sources, which would be helpful in relation to clarifying the specific sectors that need to be targeted, but it did set clear developer targets to reduce emissions:


*“Development will be expected to achieve: A minimum 10% reduction in regulated CO_2_ emissions through energy efficiency measures; and A minimum 35% reduction in regulated CO_2_ emissions through a combination of energy efficiency measures and on-site renewable energy generation”*
(Towards zero carbon development policy, LPA 1)

Regarding biodiversity protection and enhancement, there were fewer specific requirements and targets to protect species and habitats, beyond referring to national (and European) legislation. LPA 2’s policy on “Green Infrastructure and Nature Conservation” called for an unspecified percentage of “net gain” in biodiversity from developments but does require consultation of the UK Biodiversity Action Plan (BAP), Local BAP for habitats and species, and those species listed in their local Biodiversity Record Centre.

#### 3.1.2. Local Health and Wellbeing Strategies

Of the sample plans in this review, only LPA 2 and LPA 4 local plans included references to their local health and wellbeing strategies. In particular, LPA 4 required major developments to consider local health strategies as a part of their planning applications (in the “Healthy Communities” policy).

#### 3.1.3. Health Impact Assessment of Local Plans

All LPAs in the sample applied a Sustainability Appraisal (SA)/Strategic Environmental Appraisal (SEA) of the local plan policies. SA/SEA includes health principles or objectives (amongst other environment and social issues), but the principles are at quite a high strategic level, and it was unclear whether or how the SEA findings affected (increased or amended) any specific health-related requirements for developers in the local plans. LPA 4 was the only local authority to conduct an HIA of their local plan. LPA 4 applied the London Healthy Urban Development Unit’s (HUDU) HIA framework as it was cited by the Mayor of London and seen to be “minimally resource intensive” [55]. Their report clearly states the rationale for undertaking an HIA:


*“The HIA assesses the potential effects of the Local Plan policies on the health of [LPA 4]’s residents and recommends actions to mitigate any negative impacts”*
(London HUDU, 2019).

LPA 1 proposed to apply an HIA in a more complete draft of their local plan in the future, which is why they also scored green in Table 2. LPA 2 conducted a health and equalities impact assessment (HEQIA) regarding local plan requirements for developers, which updates and adds specific spatial allocation for development sites, but they did not conduct an HIA of the whole local plan document.

#### 3.1.4. Local Health Evidence

To better appraise the potential impact of a development on local health outcomes, developers need to take account of the baseline local health data and trends [56]. In relation to linking local plan priorities to local health evidence, LPA 4 was the only LPA to include a direct reference to their local Joint Strategic Needs Assessment. All the local plans referred to area-scale life expectancy differences and deprivation data in comparison with national averages, but they did not highlight ward level data, such as relating to Indices of Multiple Deprivation. This local health evidence base provides important contextual information which can help to inform and shape development proposals. Local data regarding health deprivation and other key indices of deprivation are provided nationally by the Consumer Data Research Centre [57]. LPA 7′s Joint Strategic Needs Assessment extracted local health data to present maps and summaries of ward and neighbourhood-scale data on health inequalities and deprivation levels; however, their local plan did not signpost developers to take account of this valuable local health evidence base.

#### 3.1.5. Signposting to Health Standards and National Guidance

All the local plans referred to standards that seek to, directly and indirectly, benefit health outcomes, such as referring to the nationally described space standards, as well as requiring the Building Research Establishment Environment Assessment Method (BREEAM) for commercial buildings, BREEAM Communities for major developments, or as LPA 2 propose, the “One Planet Communities” standard. LPAs 2 and 5 referred to the “Lifetime Homes” standard which promotes accessibility and adaptability for people with disabilities and special needs. No local plan in this sample referred to the “Building for a Healthy Life” (BHL) design standard [58], although LPA 3 and LPA 7 do recommend the older Building for Life version of the standard. The BHL guidance includes three place-based themes—integrated neighbourhoods; distinctive places and streets for all—and provides visual illustrations of those principles. The principles also echo some of the Bird et al. (2018) determinants of health principles which are applied in this review. However, BHL does not cover the “healthy homes” or “healthier food environments” principles and features proposed by Bird et al. (2018).

All the plans had limited references that signpost developers to Office of Health Improvement and Disparities (OHID, previously PHE) resources. Some of the local plans signpost to some other healthy design guidance documents that the partner LPA could refer to. For example, LPA 2′s local plan indicated future open space strategies should refer to Sport England’s Active Design Guide (2015, but currently being reviewed) [59] and the National Institute for Health and Clinical Excellence (NICE)’s national guidance 90: Physical Activity and the Environment (2018) [60].

### 3.2. HLP Review Area 2: Health Requirements for Developers

The following section reviews whether the local plans clearly include policies and references relating specific health requirements for developers. Table 3 gives an overview of the score for each of the attributes in this section.

#### 3.2.1. Determinants of Health Score

The coverage of the “determinants of health” features that are reported to promote health outcomes is fairly comprehensive across all seven local plans, supporting the view that they are good practice examples. However, there are certain gaps where greater clarity and detail could be provided.

Figure 3 summarises how the local plans compare regarding the five themes and 39 “determinants of health” features (see Appendix A Table A2 for full list). In general, features relating to the themes of neighbourhood design, natural and sustainable environments and healthy transport scored more highly across the seven LPAs, in comparison with the themes of healthy housing and healthier food environments.

Neighbourhood design: Five of the seven local plans included in the review scored highly in terms of healthy neighbourhood design principles. They included a number of policies that establish comprehensive spatial and neighbourhood design principles, promoting the character and distinctiveness of place, active travel, inclusion, mixed use and typologies, compactness, optimal densities and connectivity. Most of the local plans address some of the concerns regarding increasing intensification of sites and population density. For example, LPA 1′s policy on “Homes in multiple occupation” had clear requirements to ensure that the intensification of sites would consider the impacts to existing residents and infrastructure (see Appendix A Table A3).

Healthy housing: Regarding policies that relate to interior *access to natural light*, some of the local plans included a partial reference to indoor lighting in a policy regarding climate adaption. LPA 2 and LPA 6 made a clear requirement to promote access to natural light within housing policies by referring developers to the Building Research Establishment (BRE) national guidance on access to daylight [61]. LPA 4 included a clear requirement regarding the provision of dual aspect homes, noting that it offers a range of benefits, including for ventilation and access to daylight, and it had a policy to promote dual aspect accommodation (see Appendix A Table A3).

*Housing refurbishment or retrofitting* policies should seek to promote well-maintained and well-managed existing buildings, which is reported to support general health benefits as well as reduce risks of crime associated with dilapidated areas, as well as the embodied carbon benefits of retaining buildings in good condition [7,8,62,63]. All the local plans include requirements regarding the refurbishment of commercial properties, but not regarding residential properties. Some of the local plans included policies referring to conservation areas or listed buildings. For example, LPA 5 had a policy regarding supporting refurbishment and regeneration of existing buildings (see Appendix A Table A3). LPA 6 had similar policies relating to the retention of specific heritage buildings. This raises the question whether clearer policies can be established regarding promoting the refurbishment and retrofit of existing housing stock associated with regeneration projects.

In terms of policies that address *household hazards* to health, the local plans tended to refer to specific building regulations, but not in a comprehensive way. It is notable that the policies did signpost developers to all relevant building regulations that address household hazards and limited use of language that encourages developers to go beyond these basic building regulation requirements. English building regulations include: Part A: Structure; Part B: Fire safety; Part C: Site preparation and resistance to contaminants and moisture; Part D: Toxic substances; Part E: Resistance to the passage of sound; Part F: Ventilation; Part G: Sanitation, hygiene and water efficiency; Part H: Drainage and waste disposal; Part J: Combustion appliances and fuel storage systems; Part K: Protection from falling, collision and impact; Part L: Conservation of fuel and power; Part L new requirements; Part M: Access to and use of buildings; Part N: Glazing—safety in relation to impact, opening and cleaning; Part P: Electrical safety—dwellings; Part Q: Security. (*See*
Section 3.2.4
*regarding air quality (indoor), which is relevant to this theme)*.

Healthy food environments: Regarding creating healthier food environments, there is an opportunity for strengthening policy language regarding healthy food provision in school and retail environments. For example, LPA 6′s local plan referred to their commitment to promote access to healthy food in schools through an educational scheme, “the Healthy Child Quality Mark”. LPA 2′s local plan also aimed to promote access to healthy food retail outlets and access to local food growing opportunities (see Appendix A Table A3).

Natural and sustainable environments: All the local plans had a fairly clear set of policies promoting health in relation to natural and sustainable environments. One feature for potential improvement, however, relates to climate adaptation requirements to reduce people’s exposure to extreme weather. Policies typically referred to addressing flood risk and overheating, but there is less clarity about other extreme weather/climate-related risks, such as storms, droughts or extreme cold, which should be also recognised. These factors are important in relation to energy and water efficiency measures, ventilation measures, and green infrastructure (SuDS) requirements. (*See*
Section 3.2.4
*regarding air quality (outdoor), which is relevant to this theme)*.

Healthy transport: All the local plans outlined clear transport principles and requirements, including policies that promote active travel, better connectivity, safe and efficient infrastructure, compact communities, public realm improvements, recognition of cycling, walking and public transport infrastructure, which will all benefit numerous determinants of health, and therefore health outcomes. LPA 4′s “Transport Connectivity” local plan policy makes a clear recommendation in this regard to promote strategic links across the borough (see Appendix A Table A3).

#### 3.2.2. Healthy Development Policies

Four of the seven local plans included explicit healthy development policies which outline a clear expectation that developers should contribute to health outcomes; however, all the local plans contained policies that promote better quality development and amelioration of negative impacts and risks to health such as from pollution (as required by Environmental Impact Assessments). LPA 1 had a policy, “The Health Impacts of Development”, that gives a clear statement that development “*with unacceptable health impacts will not be permitted*” and required the application of Health Impact Assessments (HIAs) for major developments. LPA 4 had a strong policy that highlights the links to relevant policies across the local plan, outlining requirements that contribute to healthy communities and calls for references to local health strategies (see Appendix A Table A3).

All seven local plans required HIA to be conducted for major or large-scale developments, except LPA 3, where an HIA is recommended and not mandatory. Notably, LPA 7 required the application of an HIA for any scale of development proposal within thirty specific site allocation areas “*within or close to an area of significant deprivation*”, but not for large-scale development proposals in other locations.

#### 3.2.3. Planning Validation Checklists

Each authority publishes a validation checklist outlining the types of evidence that developers are required or recommended to produce for different types of planning applications. Notably, none of the local plans signposted developers to refer to their validation checklists, although the developer’s planning consultants are likely to consult the checklists to ensure compliance with all the local requirements. Within the validation checklist documents, LPA 7 recommends an HIA as part of a (full or outline) planning application. LPA 5′s checklist requires a “*Health and Wellbeing statement for any residential scheme in excess of 100 units or any commercial application in excess of 5000 sqm*”. The other authorities’ validation checklists did not refer to the need to produce a health statement or conduct an HIA. All checklists did, however, include a number of expected assessments and statements that are likely to address heath impacts, either directly or indirectly, which, depending on the scale and type of development, can include design and access statements, Sustainability Appraisal, energy statements, ventilation statements, daylight and sunlight surveys, environment statements, biodiversity/ecology surveys, air quality assessments, transport assessments, noise impact assessments, landscape visual impact assessments, open space assessments, flood risk assessments, heritage assessments, etc. LPA 3 also recommends an odour impact assessment.

#### 3.2.4. Partner LPA Priority Determinants of Health

Looking in more detail at the four determinants of health that the partner LPA specified as local priorities, the following were identified.

Fuel poverty and energy security: All the local plans addressed the issue through requirements regarding energy efficiency and low-carbon development (this is linked to the Planning and Energy Act 2008, Building Regulations Part L and the Future Homes Standard, which is due to come into force in 2025). LPA 2 and LPA 3 were the only local plans to directly refer to the problem of fuel poverty and energy insecurity, however. LPA 2 raised the issue through signposting their fuel poverty strategy, which is directly quoted in their local plan (see Appendix A Table A3).

Air quality (indoor and outdoor): All seven local plans had policies to avoid, minimise and mitigate the negative impacts on outdoor air quality from pollution, particularly in relation to development within Air Quality Management Areas. The local plans also referred to policies to improve air quality, such as through supporting active travel via walking, cycling and improving access to public transport, as well as expanding electric vehicle infrastructure and green infrastructure investments.

There were fewer policies that directly address indoor air quality, however. There were indirect ventilation requirements in LPA 1′s local plan regarding using energy-efficient ventilation technology and passive ventilation, but no explicit requirement to improve indoor air quality to promote health benefits in the local plan. Local plan policies can encourage developers to promote better indoor air quality. For example, LPA 2’s local plan included interior design requirements to address of the impacts of outdoor air pollution to indoor air quality.

Security of tenure: There were limited references in all seven local plans regarding developers and/or landlords ensuring that tenants have minimum guaranteed rental periods and tenancy rights. LPA 7 alone included a policy that directly promotes security of tenure for the Build to Rent market (see Appendix A Table A3).

Affordable housing: Regarding affordable housing provisions, a number of the local plans lacked clarity about who is reflected in the definition of “special” or “specialist” needs, as well as housing provisions for homeless people. Greater differentiation will help ensure that the specific needs of different vulnerable groups of people are not ignored. For example, LPA 2′s local plan included more specific attributes relating to homeless people and other specialist needs provisions.

### 3.3. HLP Review Area 3: Implementation of Health Requirements

This section of the review raises the critical question about how local plans can be written in a way that encourages developers to implement policy requirements, including those relating to health. This considers the funding, planning and management, monitoring, and community ownership attributes of implementation. An overview of the findings is presented in Table 4.

#### 3.3.1. Funding Requirements

In general, all the local plans had clear policies indicating their expectations for financial contributions from developers towards public services, infrastructure and amenities (e.g., schools and GP services) that will promote individual and community wellbeing. They included general and specific policy specifications regarding contributions, via mechanisms such as the Community Infrastructure Levy (CIL) and/or negotiated Section 106 agreement [64].

#### 3.3.2. Viability Appraisal

Some of the local plans indicated in more detail how they expect viability appraisal to be undertaken and negotiated in order to increase the transparency and accountability of the process. For example, LPA 6′s policy “Approach to development delivery and viability, planning obligations and CIL” promoted transparent viability appraisal by requiring “*robust viability evidence…with Open Book process”*. LPA 1′s local plan also made explicitly clear which priorities it expected developers to factor into viability appraisal at an early stage in the process, such as in relation to zero carbon policy:


*“At the inception of development proposals, developers should build achieving zero carbon into their consideration of scheme viability”*
(“Towards zero carbon development”, LPA 1 local plan).

#### 3.3.3. Monitoring

Monitoring needs to be explicitly required so that developers are clear about LPA expectations to keep track of delivery. It was interesting to note that all the local plans included numerous requirements for the LPAs to assess and monitor particular policy areas themselves, but none of them included direct monitoring or post-occupancy evaluation (POE) requirements for developers regarding health outcomes, e.g., requirements to fund and undertake a POE of health metrics at a given stage, once a site is in use, as well as expectations that future development phases will be refined based on the POE findings.

The local plans did include developer monitoring requirements for other policy areas that will indirectly affect health, including regarding carbon emissions, biodiversity, and water quality. For example, LPA 2′s local plan stated that developers should demonstrate their contribution to the local carbon emissions reduction standard through an energy statement before commencing a project and, importantly, to review progress post-construction.

#### 3.3.4. Planning, Management and Maintenance

As shown in Table 4, LPA 5‘s local plan was the only one that included a requirement for major developers to provide “details of ongoing [health] management or mitigation of issues where necessary”, as part of the HIA process. No other plans required developers to include a health promotion management plan, associated with the HIA or otherwise. These local plans were scored an amber rating as there are requirements in most of the local plans for developers to produce management and maintenance plans in relation to a number of issues, particularly regarding environmental areas: pollution control, biodiversity, green infrastructure, open spaces, flood risk, sustainable drainage systems (SuDS) and low or zero carbon plans.

#### 3.3.5. Community Ownership

Whilst some of the local plans referred to the self-build register of interest, they do not clearly indicate whether or how developers might be expected to engage with the register or where to find it online. Some included language about the LPA seeking to “encourage” or “support” community-led housing, development and land-trusts, but it was often unclear what practical steps they would take to do so or their expectations from developers in this regard. For large, masterplanned projects (involving a mix of residential and commercial uses and likely to be over 100 dwellings), a developer can set aside parcels of land to include self-build, cooperative or other community led models. LPA 3′s local plan actually set a minimum requirement for custom and/or self-build housing to be adopted in all new developments of “*30% affordable housing and 5% self and/or custom build housing” (*“Role and function of places policy”, LPA 3 local plan).

#### 3.3.6. Statement of Community Involvement

Regarding references to the Statement of Community Involvement (SCI), four out of the seven local plans reviewed include a reference or footnote to their SCI guidance and principles (or similar community engagement guidance). None of them indicated a clear expectation or recommendation that developers should refer to this guidance on when engaging communities as a part of a planning application or when preparing their design and access statements.

## 4. Discussion

The following discussion considers opportunities to improve health references across the seven local plans, as identified through the HLP review process. It reviews the strengths and weaknesses of the approach adopted, and identifies opportunities for future work.

### 4.1. Comparing Health References and Gaps in Different Local Plans

The HLP review identified many similarities and good policy language across all seven local plans, in terms of the health-specific requirements and policies relating to the various attributes and determinants of health included in the review framework, as presented in Section 3 and highlighted in Table 2, Table 3 and Table 4 and Figure 3. Particularly strong was the policy language adopted in relation to developer requirements that seek to promote good quality neighbourhood design and place making, as well as regarding transport infrastructure that promotes inclusive and active travel. There was also common language where there is clear national (and EU) legislation or policy guidance (e.g., as outlined in NPPF and EIA legislation), such as regarding carbon reduction targets, biodiversity improvements and national guidance regarding indoor space provisions [40,41].

#### 4.1.1. Broad Opportunities to Strengthen the Integration of Health in Local Plans

The HLP review has contributed to the identification of common areas for improvement in all seven local plans, which would enhance the integration of health. These recommendations include:(i)Specification of local health priorities: The local plans could take greater account of local health priorities, as outlined in local health and wellbeing policies and evidence, and signpost developers to these.(ii)Signposting guidance and standards: The local plans need to point developers towards national publications and voluntary standards that clarify expectations regarding good-quality development that delivers health outcomes, e.g., Building for a Healthy Life and Lifetime Homes.(iii)Clarifying specific health-related requirements for developers: The local plans should consistently incorporate policies, including cross-referencing to planning validation checklists regarding HIAs. They should specify definite health requirements regarding indoor air quality, access to daylight, fuel poverty, security of tenure, access to healthy food in schools and retail outlets and enabling public transport to recreational spaces.(iv)Strengthening implementation requirements: In order to ensure that local plan requirements better impact development practice and outcomes, they need to embed policies that encourage developers to adopt health management plans, monitoring, as well as improve opportunities for community ownership and engagement, such as through community-led housing trusts.

#### 4.1.2. LPA Type and Integration of Health in Local Plan

The overall scores for the review attributes and determinants of health assigned to each local plan, according to authority type and whether HIA was applied, do not indicate that there was a large difference in scores between the district councils and urban unitary authorities. The rural authority (LPA3) scored comparably well in terms of the attributes assessed in this review; the health requirements for developers were less clearly defined. It would be interesting to apply the review to more local plans in other urban and rural unitary authorities to see whether this difference is also the case, and if so, to examine further why that might be the case.

#### 4.1.3. Application of HIA and Integration of Health in Local Plan

In relation to the application of an HIA, of the two LPAs who had partially or fully undertaken an HIA of their local plan, LPA 2 scored highest among the seven local plans overall—which, considering their emphasis on reviewing developer impacts to health and the developer focus of their HIA, would seem to make sense. LPA 4 came fourth out of seven. This mid-range score could, in part, be linked to the fact that they applied the London HUDU HIA criteria, which does not cover all the attributes and determinants of health included in the HLP framework. For example, in terms of HLP review area 1—the health definitions attributes—HUDU does not consider policy references to local health strategies and evidence base, or references to national guidance and standards. In relation to review area 2—regarding developer requirements—HUDU covers many of the same elements but it does not refer to certain specific determinants, such as retrofitting, management and maintenance of housing; indoor air quality; or secure tenure provisions. Attributes included in review area 3—focusing on implementation—such as management plans, monitoring and funding obligations are also not considered by the HUDU framework. HUDU is designed for planning applications, so it is not surprising that it may not be entirely applicable for a local plan review. As such, one recommendation coming out of this study is to expand the HUDU HIA framework to become more relevant to local plan reviews. It could include these additional attributes described above to promote local plans that are informed by local health priorities, the latest evidence on determinants of health, and promote implementation.

Another reason that HIAs may not be applied by the other LPAs is that they will have conducted a Strategic Environmental Appraisal/Sustainability Appraisal (SEA/SA) of the local plan, which must include health elements in the policy appraisal [65]. What is unclear is precisely which aspects of health are included in their SEAs. This points to a second recommendation for greater transparency at a national level about the specific health criteria that should be considered to be relevant for an SEA, if an HIA is not conducted separately. This clarity would help both LPAs and developers to better understand those aspects of health that are important to consider as a part of planning policy, applications and decisions.

#### 4.1.4. Strength of Policy Language in Local Plans

The comparative analysis across local plans meant that the partner LPA public health and planning officers were able to contrast draft policy language with similar policies used by other LPAs. This was especially of interest where local plans have already passed the scrutiny of the national Planning Inspectorate and are formally adopted as local planning policy. The successful adoption of policies by other LPAs sets a precedent about what the government will allow and reduces the likelihood of local plans being rejected by the Planning Inspectorate where they apply similar language. As such, the comparison of policy language used in adopted local plans provided legitimisation enabling planning officers to advocate for similar policy language to their political representatives.

The comparison of policy language also highlighted variation in the detail and strength of the health requirements between the local plans. In particular, there is variation as to whether opt-out clauses were used. The term “opt-out-clause” refers to the use of language that waters down planning obligations by leaving room for planning applicants to negotiate or even avoid a policy requirement, for example, where policy requirements are clarified with terms such as “where viable” or “where appropriate”. For example, LPA 2′s local plan policy on “Sustainable Buildings” states:


*“The council will seek that all new development incorporate sustainable design features to avoid expansion of the city’s ecological footprint…Unless it can be demonstrated that doing so is not technically feasible and/or would make the scheme unviable”.*


Similarly, LPA 5′s local plan used the term “where appropriate” 18 times regarding developer requirements. An example is their policy on air pollution, which states: “*Where appropriate Major developments should incorporate measures to reduce and minimize air pollution*”. Such language creates ambiguity about how and LPA should define and measure what is meant by “appropriate” and what is deemed as essential to include or exclude from viability appraisal. This highlights the need to better understand how developers and planning consultants interpret such ambiguous policy language in practice.

It was suggested by the partner LPA that such opt-out clauses were often introduced by the Planning Inspectorate when draft plans were submitted. This points to a perception of the increased cost of health-related policy requirements coming into conflict with the need to meet financial viability and housing targets [41,66]. This raises the question of whether the Planning Inspectorate is in fact facilitating the submission of weaker proposals from developers, and whether more definite and finer-grained detail regarding policy expectations for specific determinants of health can be consistently supported at national as well as local levels. Further examination of these details will help address the third area of the HLP review regarding implementation, ensuring that local plans carry sufficient weight to ensure that urban development proposals better incorporate local health priorities.

### 4.2. Strengths and Limitations

This study indicates the benefit of undertaking a comparative review across different local plan policy documents. The co-production approach ensured that the HLP review framework was relevant to the key concerns faced by LPAs when it comes to supporting the health of their local population. This process supported the iterative and reflexive refinement of the review framework [67]. It helped ensure that the framework was better aligned with the policy language more commonly understood and applied by planning officers, and improved clarity about the definitions of review attributes and features. As a result, the review framework is more likely to have wider application and relevance to other LPAs in England [68].

The sample included in the comparative review cannot, however, be considered a representative group, but as a purposive sample, it has enabled the initial development and trialling of the HLP review framework. As such, it would be valuable to apply the framework to a broader range of local plans, with differing LPA types and rural/urban locations, to examine whether the health-related policy gaps identified in these seven local plans are indeed similar elsewhere.

### 4.3. Further Research Opportunities

Colleagues at the partner LPA provided positive indications about incorporating the recommendations of the HLP review into the next draft of their local plan. It will be important to follow up the process with an evaluation of the impact of the review. This is necessary to identify whether and how the review affects the next stage of the local plan policies—in particular, whether there are additional references to the specific health attributes identified by the review recommendations. In the longer term, it would also be valuable to examine whether there is thought to be an increased incidence in references to health commitments in the future development applications that are submitted to the LPA.

As already indicated, it would be beneficial to undertake further work to examine the wider applicability of the HLP review framework, as well as the wider relevance of the general findings from this small sample, for other English LPAs and their local plans. Furthermore, there is a need to gain greater insight into how developers, their planning consultants and other contractors, and interpret the policy language in local plans in practice. This is particularly a concern in relation to the use of “opt-out-clauses” or conditional policies which may facilitate weak implementation of requirements by applicants. Part of that work should be to examine the role of the Planning Inspectorate in facilitating the watering down of health requirements.

## 5. Conclusions

This study identifies the need for greater coherence regarding health requirements in local plans and national planning policy and guidance. It presents an explanation of the process of developing and implementing a co-produced and comparative Health in Local Plans policy review framework to enhance the inclusion of policies that will support healthier urban development.

Firstly, the HLP review framework has contributed to the identification of opportunities to enhance local plans, adopting more targeted, inclusive and healthier place-making policies, through highlighting key gaps and policy language contained in other local plans that can potentially be adapted and applied elsewhere. The findings identify specific recommendations to strengthen the consideration of Health in Local Plan policies, including (i) ensuring that policies are informed by and signpost to local health priorities; (ii) signposting to national guidance and standards; (iii) strengthening health-related requirements for developers, including indoor air quality, access to daylight, fuel poverty and security of tenure; and (iv) improving the implementation of requirements for developers, through adopting health management plans, monitoring and community ownership models.

Secondly, the framework has helped identify areas to improve the systematisation of health considerations in national policy. We suggest that this review could help enhance guidance about conducting HIAs of local plans. This includes (i) incorporating the assessment of policy references to local health priorities and national guidance, (ii) reflecting evidence on determinants of health and (iii) including language that promotes the effective implementation of policy requirements. Similarly, where SEAs/SAs are applied, LPAs are recommended to adopt greater transparency about the specific health elements that are appraised in the review of local plans.

This study would benefit from a follow-up review to appraise the impact of this process for the partner LPA and their local plan, as well as to consider the wider application of the HLP review framework with other LPAs in England to understand whether there is a more general need to strengthen specific health policies in local plans. Further research would also be beneficial seeking to understand the drivers and practical implications of opt-out clauses used in local plan policy language for developers. As such, it is hoped that this small-scale study will prove a valuable starting point to address barriers and improve the systematic integration of health attributes and features in future local plan policies.

## Figures and Tables

**Figure 1 ijerph-20-04079-f001:**
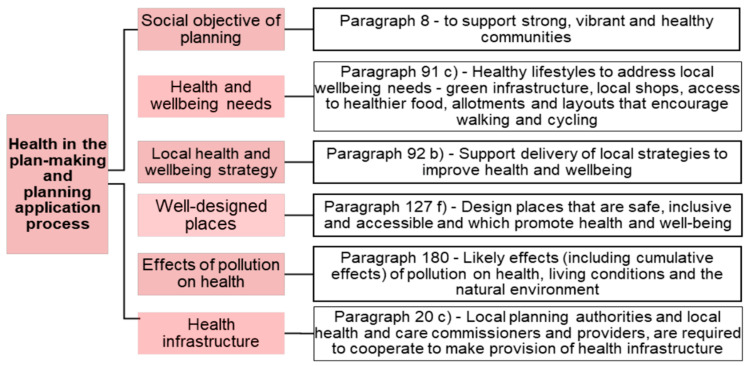
Health policy elements in the National Planning Policy Framework 2012 (source: PHE 2020 [11]).

**Figure 2 ijerph-20-04079-f002:**
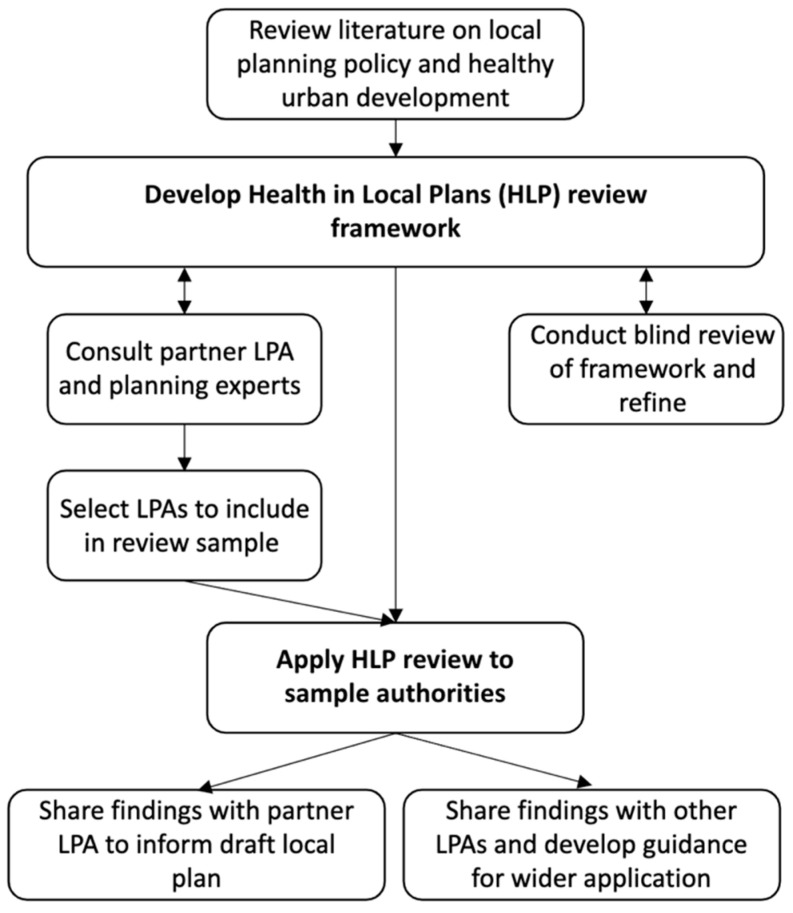
Steps to formulate the Health in Local Plans review framework.

**Figure 3 ijerph-20-04079-f003:**
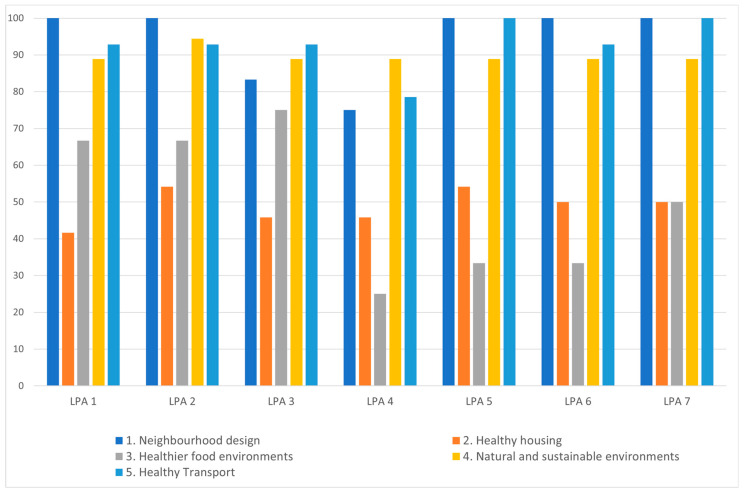
Determinants of Health percentage score for five themes, comparing seven local plans.

**Table 1 ijerph-20-04079-t001:** Local plans included in the review.

Local Plan	Council Type ^1^	Indices of Multiple Deprivation (Decile Ranking, Comparing 333 LPAs) ^2^	Date Local Plan Adopted	HIA of Local Plan
LPA 1	Unitary	3	Draft (2019)	Yes, proposed
LPA 2	Unitary	5	Part 1—2016 Part 2—2022	No (part 1) Yes, partially (part 2)
LPA 3	Unitary	3	2016 Updated 2021	No
LPA 4	Metropolitan borough	6	2021	Yes
LPA 5	Metropolitan district	1	2022	No
LPA 6	Unitary	3	2019	No
LPA 7	Metropolitan district	2	Draft	No

Key: HIA = Health Impact Assessment. ^1^ = English local planning authorities are divided into a two-tier system, with county councils and district, borough or city councils, who have slightly differing planning powers, e.g., county councils oversee planning, whilst the lower tier reviews planning applications. Some city and metropolitan boroughs are defined as unitary authorities who have combined planning powers. ^2^
*=* Indices of Multiple Deprivation (2019) is based on an average national ranking, comprising the following composite metrics: income deprivation; employment deprivation; education, skills and training; health deprivation and disabilities; crime; barriers to housing and services; living environment deprivation. The decile ranking ranges from 1 to 10, where a ranking of 1 indicates an LPA with a population with the highest levels of deprivation compared to other LPAs; a decile raking of 10 indicates a population with the lowest levels of deprivation. English indices of deprivation 2019-GOV.UK (www.gov.uk, accessed on 9 October 2022).

**Table 2 ijerph-20-04079-t002:** Local plans and definitions of health.

	Attributes	Health Definition(NCDs, Mental Health, Life Expectancy, Inequalities, Planetary Health)	Local Health & Wellbeing Strategy	HIA of Local Plan, Sustainability Appraisal	Evidence Base (JSNA, IMD)	National Standards (BfHL, Plus Other Standards, e.g., BREEAM, WELL, Building with Nature)	National Health Guidance/References
Local Plan	
**LPA1**		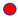				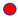
**LPA2**						
**LPA3**		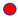				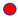
**LPA4**						
**LPA5**		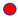				
**LPA6**		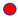				
**LPA7**		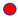				

Key: NCDs = non-communicable diseases; HIA = Health Impact Assessment; JSNA = Joint Strategic Needs Assessment; IMD = Indices of Multiple Deprivation; BfHL—Building for a Healthy Life standard; BREEAM—Building Research Establishment Environmental Assessment Method. Traffic light score: green = feature is clearly present; amber = partially present; red = not present.

**Table 3 ijerph-20-04079-t003:** Local plans and health requirements for developers.

	Attributes	Determinants of Health Score	Healthy Development Policies(HIA, Promotion, Mitigation)	Validation Check Lists (Referring to HIA)	Fuel Poverty/Energy Security	Outdoor Air Quality	Indoor Air Quality/Ventilation	Secure Tenure	Affordability Housing Provisions
Local Plan	
**LPA1**	76%(29/39)						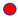	
**LPA2**	81%(31.5/39)							
**LPA3**	64%(25/39)						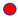	
**LPA4**	76%(29.5/39)							
**LPA5**	73%(30/39)							
**LPA6**	76%(29.5/39)					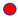		
**LPA7**	77%(30/39)					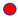		

Key: HIA = Health Impact Assessment. Traffic light score: green = feature is clearly present; amber = partially present; red = not present.

**Table 4 ijerph-20-04079-t004:** Local plan and implementation of health requirements.

	Attributes	Funding Requirements	Viability	Monitoring	Management & Maintenance Plans	Community Ownership	Statement of Community Involvement Guidance
Local Plan	
**LPA1**			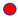			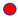
**LPA2**						
**LPA3**			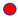			
**LPA4**			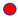			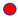
**LPA5**						
**LPA6**			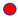			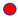
**LPA7**						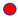

Traffic light score: green = feature is clearly present; amber = partially present; red = not present.

## Data Availability

Data available on request due to anonymity restrictions.

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
