# Peer review of "Integrating Health into Local Plans: A Comparative Review of Health Requirements for Urban Development in Seven Local Planning Authorities in England"

_ijerph, 2023, doi:10.3390/ijerph20054079_

Round 1
Reviewer 1 Report
Title
Much of the article is about health requirements for developers, which is not reflected in the title of the paper.
General comments
What I miss is a systematized description of the system for implementing health issues in documents and planning policy in the UK. The authors write about LAP, HWS, HIA etc., on p. 13 there are mentions of SCI and DAS, on p. 17 about London's HUDU and HEQIA, on p. 20 about BRE national guidance. Additionally, section 3.1.5 refers to various standards, and section 3.2.3 also contains a planning validation checklist regarding HIAs.
This lack of systematization and dispersion of information provokes several questions (the order is not apparent from the structure of the text) : 1) what is the legal framework for the implementation of health-related issues in planning and strategic documents at the national/local level; 2) what is the hierarchy - which documents apply at the national level and are implemented at the local level, 3) Which of the listed documents are mandatory for local authorities to develop, 4) how do the provisions of PHE's guidance relate to documents like LAPs and HWS, 5) does the LAP really mandatorily integrate the provisions of HWS (line 155-157), because only 2 of the analysed LAPs include them (p. 16) - why; 6) how do the NPPF 2019/2021 health provisions correspond to the scope of content of new or updated LAPs.
With regard to developers, it is not clear whether they are obliged (under current law or the provisions of planning and strategic documents at various levels) to include health issues in their investments? To what extent does this follow from legal framework and to what extent from local level regulations?
I suggest adding a section on the implementation of health issues in legislation and strategic-planning documents and regulations on standards and validation in the UK supplemented by a compilation of requirements for developers (national/local level). Additional graphics will be a useful supplement to the content.
It is not evident from the text what is novel about this study, what is the contribution of this research to the state of the art. To be supplemented.
Section 2.3 (p.13)
The sampling is not clearly described. It is not demonstrated why these and not other LAPs were analysed.
"Predominantly unitary or metropolitan borough councils were included to align with BCC which is a unitary metropolitan council" Why?
In addition, Table 3-4 shows only the names of LAPs 1-7, while Table 5 shows the names of cities from Bristol to Wakefield. Does LAP 1 refer to Bristol?
The discussion doesn't really bring any new information just mostly duplicates info from Results + a few quotes from the analysed LAPs referring to different wording and clauses + information that "This is especially of interest where local plans have already passed the scrutiny of the national Planning Inspectorate" - why is this important?
The conclusions are largely a repetition of the information from the abstract. I suggest expanding the scope of this section.
Author Response
|
Comments |
Response |
|
Title Much of the article is about health requirements for developers, which is not reflected in the title of the paper.
|
New title: Integrating health into local plans: A comparative review of health requirements for urban development in seven local planning authorities in England |
|
General comments What I miss is a systematized description of the system for implementing health issues in documents and planning policy in the UK. The authors write about LAP, HWS, HIA etc., on p. 13 there are mentions of SCI and DAS, on p. 17 about London's HUDU and HEQIA, on p. 20 about BRE national guidance. Additionally, section 3.1.5 refers to various standards, and section 3.2.3 also contains a planning validation checklist regarding HIAs. This lack of systematization and dispersion of information provokes several questions (the order is not apparent from the structure of the text) : 1) what is the legal framework for the implementation of health-related issues in planning and strategic documents at the national/local level; 2) what is the hierarchy - which documents apply at the national level and are implemented at the local level, 3) Which of the listed documents are mandatory for local authorities to develop, 4) how do the provisions of PHE's guidance relate to documents like LAPs and HWS, 5) does the LAP really mandatorily integrate the provisions of HWS (line 155-157), because only 2 of the analysed LAPs include them (p. 16) - why; 6) how do the NPPF 2019/2021 health provisions correspond to the scope of content of new or updated LAPs. |
The reviewer identifies a fundamental issue in the English planning system, that there is a lack of systematisation in relation to health policy and where it sits in the planning system – which makes presenting a systematised description of the system somewhat problematic. It is an important point and we have added additional text to highlight this problem and give a clearer overview of the various key policies and institutions that seek to address health in planning.
New paras and amendments have been added (lines 46-154) to point out the present the context and inconsistencies regarding health in planning policy, including: I. The national and local policy context, including further details about the health focus in the NPPF, PHE guidance, HWS, HIAs, SCIs and links to Local plans. A new Figure 1 has been added indicating the key health elements in the NPPF that also clarifies this policy context. 2. Hierarchy is clarified through depicting the national and local regulatory and guidance context, 3. As per point 2, clearer wording has been included about whether a policy or guidance is voluntary or mandatory 4. Additional wording about PHE’s role and guidance, particularly in relation to HIAs ((lines 75-85). Also new text Health and Wellbeing strategies (lines 59-60) 5. The NPPF does not require for local plans to integrate HWS provisions / priorities (see lines 63-66) 6. Further details about NPPF provisions are included regarding LPA requirements (lines 55-70)
|
|
With regard to developers, it is not clear whether they are obliged (under current law or the provisions of planning and strategic documents at various levels) to include health issues in their investments? To what extent does this follow from legal framework and to what extent from local level regulations? I suggest adding a section on the implementation of health issues in legislation and strategic-planning documents and regulations on standards and validation in the UK supplemented by a compilation of requirements for developers (national/local level).
|
New text and amendments have been added to clarify this relationship (lines 87-113), including: “When preparing development proposals and planning applications developers are expected to incorporate an array of national and local regulatory and planning policy requirements. Nationally, this includes the NPPF (currently under review) and other legislation, such as the Localism Act 2011, and The Town and Country Planning (Environmental Impact Assessment) Regulations 2017). They must also address local planning requirements and guidelines (as outlined in the local plan and supplementary planning documents) (5). A review of the planning system by the Town and Country Planning Association (2018) stated that planning “has undergone a bewildering rate of change and is now fragmented and confusing.” (13). Similarly, Barton and Grant (2013) noted: “There have been recurrent calls for the planning system to employ new methods that will challenge well-established ways of working and better integrate the evaluation of potential outcomes of planning for more sustainable development”(19). Within this fragmented context, multiple studies have pointed to a lack of systematic approach in the planning system towards health (4–6).” |
|
Additional graphics will be a useful supplement to the content. |
Figure 1 has been added to help clarify the national planning context |
|
It is not evident from the text what is novel about this study, what is the contribution of this research to the state of the art. To be supplemented.
|
Added to intro (lines 146-154): “The framework seeks to address the lack of systematisation of health requirements in local and national planning policies, through identifying specific evidence-based attributes and determinants of health that should be addressed through local plans. The work contributes to field by establishing a more coherent approach to considering health outcomes in future development proposals submitted and delivered within an LPA area. Learning from this study can support future application of the HLP review framework in the drafting of other local plans across England, and potentially more widely, to support finer grain integration of health priorities.”
|
|
Section 2.3 (p.13) The sampling is not clearly described. It is not demonstrated why these and not other LAPs were analysed.
“Predominantly unitary or metropolitan borough councils were included to align with BCC which is a unitary metropolitan council” Why?
In addition, Table 3-4 shows only the names of LAPs 1-7, while Table 5 shows the names of cities from Bristol to Wakefield. Does LAP 1 refer to Bristol?
|
More detail added about the sampling (lines 377-437) to explain the rationale for including the four selection criteria
Amended text (lines 421-426) to clarify why predominantly urban unitary and metropolitan authorities were selected to support better comparability with the partner LPA in terms of planning functions (reference to BCC amended to partner LPA to anonymise).
Amended Table 5 (now Table 4) to anonymise LPAs consistently throughout the paper. It was decided to anonymise the sample authorities to avoid any negative connotations from the critique of a particular authority. We’re looking to co-produce guidance as the next step in this project and don’t want to risk relationship issues with LPAs by identifying them. We do not feel that highlighting that their identities is central to the messaging and findings in the paper.
|
|
The discussion doesn’t really bring any new information just mostly duplicates info from Results + a few quotes from the analysed LAPs referring to different wording and clauses + information that
|
Separate sub-headings, additional discussion and amendments have been added that looks at whether the results identified any difference between different types of LPAs and whether the LPA applied an HIA had an effect on their score. (See lines 848-1012) |
|
"This is especially of interest where local plans have already passed the scrutiny of the national Planning Inspectorate" - why is this important? |
New text added to clarify the importance of local plans being accepted by the planning inspectorate before they can be formally adopted as planning policy (lines 933`-941):
“This study indicates the benefit of undertaking a comparative review across different local plan policy documents. The comparative analysis meant that the partner LPA public health and planning officers were able to contrast draft policy language with similar policies used by other LPA. This was especially of interest where local plans have already passed the scrutiny of the national Planning Inspectorate and are formally adopted as local planning policy. The successful adoption of policies by other LPAs sets a precedent about what the government will allow and reduces the likelihood of local plans being rejected by the planning inspectorate where they apply similar language. As such, the comparison of policy language used in adopted local plans provided legitimization enabling planning officers to advocate for similar policy language to their political representatives.”
See also the introduction, explaining the role of the Planning Inspectorate
|
|
The conclusions are largely a repetition of the information from the abstract. I suggest expanding the scope of this section.
|
Additional text and amendments have been made to both the abstract and conclusion to make these sections less repetitive, and to clarify the local and national findings identified by the study. In particular the conclusion now includes more detail than in the abstract. |

Reviewer 2 Report
Dear Authors,
I am impressed about your engagement with the topic and appreciate your profound knowledge on transdisciplinary research which both is reflected in the manuscript. In the following, please find my comments and recommendations.
Title:
Please add the geographical context of your work in the subheading.
Introduction:
Please, insert a paragraph in order to place your work to a broad context and explain the need for dealing with health issues in spatial or rather urban planning and the relevance of spatial planning for health protection and health promotion.
Please, provide information on the variety of authorities and stakeholders (developers!) and their roles in local/urban planning in England.
General remark: Subheadings may be useful to structure the content.
Methods:
Please, provide information on the study context.
In order to increase readability, please, shift the extensive tables to the annex and provide concise paragraphs on your review framework as well as on Bird’s et al.s principles.
A flow chart of procedure helps to show the complexity of your efforts.
Results:
This chapter (line 453ff) contains comprehensive information and a range of text boxes with extensive direct quotations addressing particular local plans (reference sources/in-text citations are missing). Please, shift the text boxes to the annex in order to increase readability.
Direct Quotations (cf. line 453ff): please, add reference sources.
Table 4, first column: Please, provide the name of the city or rather metropolitan borough (cf. Table 5).
Line 410: typing error (Table 4).
Line 712: error in table numbering. Moreover, please, provide the name of the city or rather metropolitan borough (cf. Table 5).
All the best!
Author Response
|
Comments |
Response |
|
Title: Please add the geographical context of your work in the subheading.
|
New title: Integrating health into local plans: A comparative review of health requirements for urban development in seven local planning authorities in England
|
|
Introduction: Please, insert a paragraph in order to place your work to a broad context and explain the need for dealing with health issues in spatial or rather urban planning and the relevance of spatial planning for health protection and health promotion. |
New paras and amendments have been added (lines 46-154) to point out the inconsistencies regarding health in UK planning policy, expanding the detail about the national and local policy context regarding health obligations and voluntary guidance. Further details about the health focus of the NPPF, PHE guidance, HWS and links to Local plans are added. A new Figure 1 has been added indicating the key health elements in the NPPF that establishes the regulatory context and hierarchy. |
|
Please, provide information on the variety of authorities and stakeholders (developers!) and their roles in local/urban planning in England.
|
Added text on the different types and planning responsibilities of local planning authorities (lines 46-52) |
|
General remark: Subheadings may be useful to structure the content.
|
Added new subheadings to provide more structure to the methods, results and discussion |
|
Methods: Please, provide information on the study context. |
The study context is outlined in the introduction regarding local plans and the methods under 2.1 regarding the co-production with the partner LPA. |
|
In order to increase readability, please, shift the extensive tables to the annex and provide concise paragraphs on your review framework as well as on Bird’s et al.s principles.
|
These tables have been moved to the appendix 1 and 2 for improved readability.
|
|
A flow chart of procedure helps to show the complexity of your efforts.
|
Figure 2 has been added to outline the steps in volved in the process |
|
This chapter (line 453ff) contains comprehensive information and a range of text boxes with extensive direct quotations addressing particular local plans (reference sources/in-text citations are missing). Please, shift the text boxes to the annex in order to increase readability.
|
Most of the quotes have now been moved to the Appendix 3 to improve the flow of the text.
|
|
Direct Quotations (cf. line 453ff): please, add reference sources.
|
Reference has been added |
|
Table 4, first column: Please, provide the name of the city or rather metropolitan borough (cf. Table 5).
|
All LPAs should have been anonymised for this paper. Table 5 (now table 4) has been updated to anonymise them. It was decided to anonymise the sample authorities to avoid any negative connotations from the critique to a particular authority. We're looking to co-produce guidance as the next step in this project and don't want to risk relationship issues with LPAs by identifying them. We do not feel that highlighting that their identities is central to the messaging and findings in the paper |
|
Line 410: typing error (Table 4).
|
Amended text (now table 3) |
|
Results: Line 712: error in table numbering. Moreover, please, provide the name of the city or rather metropolitan borough (cf. Table 5).
|
As above, the LPAs have been anonymised but we have indicated in the discussion whether an LPA is a unitary, district or metropolitan borough
|

Round 2
Reviewer 1 Report
The authors have taken into account all the reviewer's comments. As a result, the issues discussed were presented and analyzed in a comprehensive and complete manner. The problem of lack of systematization with regard to health policy and its position in the planning system was also highlighted. After reorganization and the addition of new content and figures, the text provides a valuable set of analyses and recommendations for the development of local plans with an emphasis on better integration of health priorities, as well as guidance on strengthening health requirements for developers.